# Current Possibilities of Gynecologic Cancer Treatment with the Use of Immune Checkpoint Inhibitors

**DOI:** 10.3390/ijms20194705

**Published:** 2019-09-23

**Authors:** Ewelina Grywalska, Małgorzata Sobstyl, Lechosław Putowski, Jacek Roliński

**Affiliations:** 1Department of Clinical Immunology and Immunotherapy, Medical University of Lublin, 4a Chodzki Street, 20-093 Lublin, Poland; jacek.rolinski@gmail.com; 2Department of Clinical Immunology, St. John’s Cancer Centre, 7 Jaczewskiego Street, 20-090 Lublin, Poland; 3Chair and Department of Gynaecology and Gynaecological Endocrinology, Medical University of Lublin, Aleje Racławickie 23 (SPSW), 20-037 Lublin, Poland; sobma@poczta.onet.pl (M.S.); putowskil@yahoo.com (L.P.)

**Keywords:** cytotoxic T-lymphocyte-associated antigen-4, endometrial cancer, ovarian cancer, programmed cell death protein 1, T cell exhaustion

## Abstract

Despite the ongoing progress in cancer research, the global cancer burden has increased to 18.1 million new cases and 9.6 million deaths in 2018. Gynecological cancers, such as ovarian, endometrial, and cervical cancers, considerably contribute to global cancer burden, leading to $5,862.6, $2,945.7, and $1,543.9 million of annual costs of cancer care, respectively. Thus, the development of effective therapies against gynecological cancers is still a largely unmet medical need. One of the novel therapeutic approaches is to induce anti-cancer immunity by the inhibition of the immune checkpoint pathways using monoclonal antibodies. The molecular targets for monoclonal antibodies are cytotoxic T lymphocyte-associated protein-4 (CTLA-4), programmed cell death protein-1 (PD-1), and programmed death-ligand 1 (PD-L1). The rationale for the use of immune checkpoint inhibitors in patients with gynecological cancers was based on the immunohistological studies showing high expression levels of PD-1 and PD-L1 in those cancers. Currently available immune checkpoint inhibitors include nivolumab, pembrolizumab, atezolizumab, avelumab, durvalumab, and ipilimumab. The efficacy and safety of these inhibitors, used as monotherapy and with combination with chemotherapy, is being currently evaluated in several clinical studies. As the results are promising, more clinical trials are being planned, which may lead to the development of efficient therapies for gynecological cancer patients.

## 1. Introduction

Despite the ongoing progress in cancer research, the health and economic burden of cancer is increasing worldwide, leading to premature death and disability [1]. In 2018, the global cancer burden increased to 18.1 million new cases and 9.6 million deaths [2]. Between 2018 and 2040, the number of new cases is predicted to grow by over 60% to 29.4 million, and the number of cancer deaths will increase by 70% to 16.3 million [2].

Gynecological cancers, such as cervical, ovarian, and endometrial cancers, contribute considerably to the global cancer burden. In 2012, cervical cancer was the fourth most frequently diagnosed cancer among women worldwide, with an estimated 527,600 cases, and the fourth leading cause of cancer death with 265,700 deaths [3]. In developing countries, cervical cancer is the second most commonly diagnosed cancer after breast cancer, and the third leading cause of cancer death after breast and lung cancers [3]. Currently, more than 569,847 women are diagnosed with cervical cancer annually worldwide, resulting in over 311,365 deaths [4]. In 2012, endometrial cancer was the sixth most common cancer among women worldwide and the 14th leading cause of cancer-related death, with an estimated 319,600 cases and 76,200 deaths [3]. In 2012, ovarian cancer was the seventh most common cancer, and eighth most common cause of death from cancer in women worldwide, with 239,000 cases and 152,000 deaths [3]. Almost 600,000 women live within five years after an ovarian cancer diagnosis (five year prevalence). The Globocan study predicts that by 2035, the incidence and number of deaths due to ovarian cancer will increase by 55% to 371,000 and 67% to 254,000, respectively [3]. In 2018, the estimated costs of cancer care in the U.S. were $5862.6 million for ovarian cancer, $2945.7 million for endometrial cancer, and $1543.9 million for cervical cancer [5]. With an aging population, the proportion of women over the age of 65 with cancer is expected to rise substantially over the next decade [6,7]. Thus, the development of effective therapies against gynecological cancers is still a large unmet medical need [8,9].

The growth and progression of cancer is associated with evading of the anti-cancer immune response. To do so, cancer cells activate different suppressive immune checkpoint pathways that play a key role in regulating the immune system by maintaining self-tolerance and preventing autoimmunity [10]. As a result, the ability of the immune system to mount an effective anti-tumor response is restrained. Therefore, the inhibition of the immune checkpoint pathways is a promising therapeutic approach for inducing effective anti-cancer immunity. Because many of these pathways are initiated by ligand–receptor interactions on the surface of immune cells, such interactions can be targeted by monoclonal antibodies.

In this manuscript, we review the use of currently available immune checkpoint inhibitors in cervical, endometrial, and ovarian cancers.

### 1.1. The CTLA-4 and PD-1 Immune Checkpoints

The first clinically targeted immune checkpoint receptors were cytotoxic T lymphocyte-associated protein-4 (CTLA-4) and programmed cell death protein-1 (PD-1) [11]. CTLA-4 and PD-1 belong to the same CD28 family of T cell receptors and act as negative regulators attenuating normal T cell activation to prevent pathologic over-activation (Figure 1). CTLA-4 is expressed on the surface of naive effector T cells and regulatory T cells (Tregs) [12,13]. CTLA-4 functions primarily in secondary lymphoid organs, where stimulation of naive T cells induces upregulation of CTLA-4 and its competition with CD28 for B7, resulting in suppression of T cell activity [14]. PD-1 is expressed on the surface of activated T cells, Tregs, activated B cells, and NK cells [15]. PD-1 functions primarily in peripheral tissues, where T cells may encounter the immunosuppressive PD-1 ligands (programmed death-ligand 1 and 2 (PD-L1/PD-L2)) expressed by tumor cells [16,17].

### 1.2. Immune Checkpoint Inhibitors Targeting CTLA-4, PD-1, and PD-L1

CTLA-4, on the basis of its role as a negative regulator of T cell activation, has become an attractive target for therapies aiming to enhance the effector activity of T lymphocytes. The first, and currently the only, U.S. Food and Drug Administration (FDA)-approved immune checkpoint inhibitor targeting CTLA-4 is ipilimumab—fully human IgG1κ anti-CTLA-4 monoclonal antibody [18]. Ipilimumab, marketed as Yervoy by Bristol-Myers Squibb, obtained FDA approval in 2011. This approval marked the beginning of a new era for cancer immunotherapy. Data from a pooled meta-analysis from 10 prospective and 2 retrospective studies assessing long-term survival of 1861 advanced melanoma patients have been used to estimate a 3 year survival rate of 22% for patients receiving ipilimumab therapy, with survival in some patients extending close to 10 years [19]. Although ipilimumab is currently approved only for the treatment of melanoma as monotherapy, it is also investigated as a treatment for a variety of cancer types, including renal cell carcinoma (RCC), non-small-cell lung carcinoma (NSCLC), and prostate cancer. Despite small clinical benefits against these cancers, modest improvements in patient survival have been observed in particular subsets of non-melanoma cancer patients [20].

Another fully human monoclonal antibody targeting CTLA-4 is tremelimumab—IgG2 anti-CTLA-4 monoclonal antibody marketed by AstraZeneca. Currently, tremelimumab is being investigated in clinical trials. Although to date it has not improved patient survival as monotherapy in any trials, tremelimumab is currently being investigated in combination with other checkpoint inhibitors or other regimens to assess whether it will have greater efficacy as part of combinatorial regimens [21].

The first FDA-approved (2014) immune checkpoint inhibitor targeting PD-1 is nivolumab, a fully human IgG4κ monoclonal antibody, marketed as Opdivo by Bristol-Myers Squibb. Since its approval, nivolumab has become the first-line therapy for previously untreated melanoma without an additional BRAF mutation, following a phase III trial in patients receiving nivolumab versus dacarbazine. This study documented improved objective response (OS) rate (40.0% in the case of nivolumab versus 13.9% in the case of dacarbazine), progression-free survival (PSF) (5.1 months versus 2.2 months), and overall survival (OS) (72.9% versus 42.1% at 1 year) [22]. Apart from melanoma treatment, nivolumab has been shown to exhibit positive therapeutic effects against advanced RCC, advanced squamous-cell lung cancer (SCLC) and NSCLC, and recurrent squamous-cell carcinoma of the head and neck (SCCHN) [23,24,25,26]. Additionally, significant objective response rates in patients from phase II trials have led to the approval of nivolumab as a treatment of advanced hepatocellular carcinoma (HCC), advanced urothelial carcinoma, or microsatellite instability-high colorectal cancer [27,28,29]. Moreover, the results of two independent phase I/II trials showed a combined objective response rate of 65% in nivolumab treated patients with classical Hodgkin lymphoma that resulted in the approval of nivolumab as the first checkpoint blockade inhibitor for the treatment of a hematological malignancy [30,31].

In 2014, another anti-PD-1 immune checkpoint inhibitor granted accelerated approval—pembrolizumab—a humanized IgG4κ monoclonal antibody marketed by Merck as Keytruda. Pembrolizumab was approved as an alternative to nivolumab for second-line treatment of patients with unresectable or metastatic melanoma where the disease had progressed after ipilimumab ± BRAF inhibitor therapy. In 2015, after two trials confirmed the survival benefits, pembrolizumab received expanded approval as frontline therapy for unresectable or metastatic melanoma [32,33]. The results from the phase III KEYNOTE-006 trial have proven the survival benefits of pembrolizumab therapy, showing 2 year overall survival rates of 55% in patients treated with pembrolizumab versus 43% in patients treated with ipilimumab [34]. Pembrolizumab has also been approved as a therapy for the treatment of metastatic NSCLC, after phase II/III trials reported improved progression-free and overall survival of patients with PD-L1+ tumors who received pembrolizumab compared to either docetaxel or platinum-based chemotherapy [35,36]. Other trials in the KEYNOTE series have also led to pembrolizumab’s accelerated or full approval for treating various indications of classical Hodgkin lymphoma, HNSCC, urothelial carcinoma, and gastric/gastroesophageal junction adenocarcinoma [37,38,39,40].

Apart from the blockade of PD-1 receptors on T lymphocytes, the disruption of PD-1 immune checkpoint pathway by targeting of PD-L1 has also proven to be a promising approach for improving the effector activity of anti-tumor T cells. The first anti-PD-L1 monoclonal antibody to be approved for checkpoint blockade therapy was atezolizumab (fully-humanized IgG1κ monoclonal antibody) marketed as Tecentriq by Roche Genentech’s. Despite no survival benefit for patients with advanced urothelial carcinoma of second-line atezolizumab over chemotherapy, atezolizumab caused fewer and less severe treatment-related adverse events than chemotherapy in these patients [41]. In the case of NSCLC, atezolizumab was superior over chemotherapy, resulting in higher overall survival rates than those achieved with docetaxel [42].

Other approved PD-L1 inhibitors are avelumab, the fully human IgG1λ monoclonal antibody marketed as Bavencio by EMD Sorono, Inc./Pfizer, and durvalumab, the fully human IgG1κ monoclonal antibody marketed as Imfinzi by AstraZeneca/MedImmune. On the basis of the results from a phase I/II study, avelumab became the first FDA approved treatment for metastatic Merkel cell carcinoma and received accelerated approval for the treatment of metastatic urothelial carcinoma [43,44]. Durvalumab also received accelerated approval as a second-line treatment for progressive metastatic urothelial carcinoma. The results for phase III PACIFIC trial led to its full approval for treatment of stage III NSCLC that has not progressed following concurrent chemoradiotherapy [45].

## 2. Mechanism of Action of Currently Available Immune Checkpoint Inhibitors in Cervical, Ovarian, and Endometrial Cancers

The rationale for the use of immune checkpoint inhibitors as a promising alternative to conventional cytotoxic agents in patients with gynecological cancers was, among others, based on the immunohistological studies of the expression levels of PD-1 and PD-L1. Vanderstraeten et al. estimated the expression level of PD-L1 in primary, recurrent, and metastatic endometrial cancers to be 67–100% [46]. Herzog et al., during the 2015 annual meeting of the Society of Gynecologic Oncology, reported that the highest PD-1 expression rates among studied cancer types were in endometrial cancer (75.2%), epithelial ovarian cancer (66.9%), and cervical cancer (63.1%), and the highest PD-L1 expression rates were in ovarian sex cord-stromal tumors (75.0%), uterine sarcoma (46.3%), and endometrial cancer (25.2%) [47]. Today, there are many ongoing clinical trials investigating the use of currently available immune checkpoint inhibitors in gynecologic cancers (Table 1). Immune checkpoint inhibitors are investigated both as single agents and in combination with cytotoxic chemotherapy, radiation, targeted therapies, and other immunomodulators [48,49,50].

### 2.1. Cervical Cancer

Nivolumab: An ongoing phase II clinical trial (NCT02257528) is currently evaluating the efficacy of nivolumab in treating patients with persistent, recurrent, and metastatic cervical cancer. During this study, the overall survival, frequency of objective tumor responses, the incidence of adverse events, and progression-free survival will be measured to assess antitumor activity. Obtained results show that nivolumab exhibited low antitumor activity and an acceptable safety profile in patients with persistent or recurrent cervical cancer that was previously treated with platinum-based chemotherapy. In another phase II trial (NCT02465060), treatments targeting particular genetic abnormalities (such as mutations or translocations) in tumors were investigated. In this study, patients with various tumors, including cervical cancer, and loss of MLH1 or MSH2 (a mismatch repair deficiency) were treated with nivolumab. Moreover, three ongoing clinical trials are currently evaluating the efficacy of combining nivolumab with standard chemoradiotherapy and other therapies, such as HPVST cells, INCAGN01876, and INCAGN01949 (NCT03298893, NCT03527265, NCT02379520, NCT03126110, and NCT03241173).

Pembrolizumab obtained FDA approval in June 2018 as a therapy for patients with recurrent or metastatic cervical cancer. Several ongoing studies evaluate the wide application of pembrolizumab in cervical cancer. A trial involving patients with multiple types of advanced solid tumors (NCT02628067), including cervical cancer, will evaluate the objective response rate of subjects and explore predictive biomarkers. Many clinical trials are studying the efficacy of pembrolizumab in combination with other therapies, such as chemotherapy, radiation, and as a vaccination to treat patients with cervical cancer (NCT03367871, NCT03635567, NCT3444376, NCT02635360, NCT03144466, and NCT03192059).

Atezolizumab has been tested in many clinical trials to evaluate its efficacy and safety in patients with cervical cancers. In these trials, atezolizumab was used in combination with chemotherapy, radiation, bevacizumab, and Vigil (NCT03614949, NCT03340376, NCT02921269, NCT03073525, and NCT02914470).

Avelumab: The efficacy of avelumab for treating patients with cervical cancer is being assessed in two ongoing clinical trials (NCT03260023 and NCT03217747).

Durvalumab: The efficacy and safety of durvalumab in combination with radiotherapy, ADXS11-001, tremelimumab, and Vigil were investigated in clinical trials involving subjects with cervical cancer (NCT03452332, NCT02291055, NCT01975831, and NCT02725489).

Ipilimumab: Currently, a phase I trial of ipilimumab is undergoing in patients with locally advanced cervical cancer (NCT01711515) after chemoradiation (cisplatin 6 weeks treatment and radiation). Preliminary data suggest that this immunotherapy is tolerable and shows possible activity in the studied population with a historical dismal prognosis with standard therapy [51].

#### 2.1.1. Combination Therapies

Currently, a few clinical studies are ongoing with the aim to assess the efficacy of combination therapies in patients with cervical cancer: PAPAYA study (NCT03144466) concerning the use of pembrolizumab/RT/cisplatin, NRGGY017 study concerning the use of atezolizumab and chemoradiotherapy, and NCT02921269 study concerning the use of atezolizumab and bevacizumab.

#### 2.1.2. Future Possibilities: Vaccines

Adoptive T cell therapy is a promising therapeutic option for patients with metastatic cervical cancer. In clinical trial NCT0158542, nine patients with metastatic cervical cancer after platinum-based chemotherapy or chemoradiotherapy obtained a single infusion of tumor-infiltrating T cells selected for human papillomavirus (HPV) oncoprotein E6 and E7 reactivity. Complete regression was found in two patients at 22 and 15 months after treatment, and partial regression was found in one patient 3 months after treatment [52].

ADXS11-001 is a vaccine based on genetically modified bacteria *Listeria monocytogenes* to induce the immunological response against E7 oncoprotein of HPV [53]. Promising pre-clinical studies on animal cancer models resulted in the design of the study assessing the efficacy of ADXS11-001 against cervical cancer, nasopharyngeal cancer, and anal cancer associated with HPV in humans. In 2009 Maciag et.al showed the results of a phase I clinical study assessing the safety of ADXS11-001 in 15 patients with previously treated malignant, resistant, or recurrent cervical cancer. At the end of the study, two patients died, five patients showed progression, seven patients had stable disease, and one patient showed complete remission [54]. Another clinical study where ADXS11-001 was assessed was a phase II study stage 1. In this study, patients with advanced, malignant, or recurrent cervical cancer, with the inefficient first line of systemic-dose chemotherapy, were involved. Immunotherapy was applied to 26 patients. Adverse events in grade 1–2 occurred in 91% of patients. In 38% of patients, these adverse events were associated with the vaccine and most often occurred as vomiting, nausea, chills, fatigue, and fever. Median OS was 7.7 months (95% CI: 3.9–12.4) and median PFS was 3.1 months (95% CI: 2.8–3.7). The II stage phase II clinical trials are ongoing [55]. Basu et al. published results of the phase II study assessing the efficacy and safety of ADXS11-001 in patients with recurrent cervical cancer after chemotherapy and/or radiotherapy. The 109 patients were divided into two groups, and ADXS11-001 was administered with or without cisplatin. Median PFS was 6.10 vs. 6.08 months, ORR was 17.1% vs. 14.7%. Median OS was similar for both groups—8.28 vs. 8.78 months. Adverse events were reported as mild and moderate but more in the group with cisplatin (275 vs. 429) [56].

VGX-3100 is a therapeutic synthetic DNA vaccine targeting HPV subtype 16, 18 E6, and E7 proteins. Currently, the REVAL 1 study has finished, which was a randomized, double-blind, placebo-controlled phase 2b trial involving 198 patients with cervical intraepithelial neoplasia 2/3. Of the patients, 167 received a vaccine and 42 received a placebo. The study showed histopathological regression of lesion in 49.5% of 107 patients with VGX-3100 and 30.6% of placebo. Currently, the registration of patients for phase III clinical study is in progress (REVEAL 2) [57].

### 2.2. Ovarian Cancer

Nivolumab A phase II clinical trial of nivolumab in patients with platinum-resistant recurrent ovarian cancer was carried out, demonstrating encouraging clinical efficacy and tolerability [58]. In this study, 20 patients with platinum-resistant ovarian cancer were involved and treated either with 1 or 3 mg/kg nivolumab every 2 weeks until progression or up to 48 weeks. The best overall response was 15% (95% CI: 3.2–37.9), with median PFS and OS at 3.5 and 20 months, respectively. In eight patients (20%) grade 3 or 4 adverse events occurred and two experienced severe adverse events. Four patients experienced prolonged disease control.

Pembrolizumab A non-randomized multi-cohort phase Ib clinical trial (KEYNOTE-028, NCT02054806) of pembrolizumab in patients with recurrent ovarian cancers showed that pembrolizumab is well tolerated and has antitumor activity [59]. Eligibility requirements included expression of PD-L1 in 1% of tumor nests or PD-L1 expression in the stroma. Pembrolizumab at a dose of 10 mg/kg was administrated every 2 weeks until progression, intolerable adverse effects, or for up to 2 years. Twenty-six patients were treated. The best overall response was 11.5% (95% CI: 2.4-30.2), and 23.1% of patients had evidence of tumor reduction, with 11.5% of patients having a tumor reduction of at least 30%.

Regarding ipilimumab, the first anti-cancer effects of checkpoint inhibitors in patients with IV stage ovarian cancer were shown by Hodi et al [60]. In their study, a single infusion of ipilimumab (3 mg/kg) in two stage IV ovarian cancer patients previously vaccinated with granulocyte-macrophage colony-stimulating factor modified irradiated autologous tumor cells (GVAX) was well tolerated, and decreased or stabilized the CA-125 (cancer antigen 125) levels of several months’ duration. Ipilimumab was also investigated in a phase II study in patients with recurrent platinum-sensitive ovarian cancer (NCT01611558). In this study, 40 patients were treated with 10 mg/kg ipilimumab every 3 weeks × 4 doses (induction phase) followed by 10 mg/kg every 12 weeks until progression or unacceptable toxicity. Thirty eight patients (95%) did not complete the induction phase because of disease progression (14, 35%), drug toxicity (17, 42.5%), death (1, 2.5%), or other/unreported (6, 15%). Twenty patients (50%) experienced drug-related adverse events of grade 3 or higher. The ORR was 10.3%.

The efficacy of avelumab was investigated in a phase Ib study (NCT01772004) that included 124 patients with refractory or recurrent ovarian cancer [61]. Patients were treated with 10 mg/kg every 2 weeks until progression or unacceptable toxicity. The median duration of treatment was 12 weeks. In 6.4% of the patient grade, ¾ adverse events occurred, and 8.1% of patients discontinued treatment secondary to an adverse event. Partial response for an ORR of 9.7% was experienced by 12 patients. PDL1 expressivity was determined in 74 subjects, with 57 (77.0%) of the 74 being PDL1-positive, resulting in an ORR of 12.3% in the PDL1-positive tumors compared with 5.9% in PDL1-negative tumors. These findings further add to the evidence that PDL1 expressivity may be a predictive biomarker for use of immune checkpoint inhibitors in ovarian cancer.

#### 2.2.1. Combination Therapies

The promising results of studies showing inhibition of the single immune checkpoint (monotherapy) in many solid cancers (melanoma or lung cancer, among others) did not provide substantial progress in the treatment of ovarian cancer [62,63,64]. Today, several clinical trials are conducted that combine classical chemotherapy, PARP inhibitors, and other therapies with immune checkpoint inhibitors. In the ongoing phase II study, the double immune checkpoint blockade was used—PD-1 inhibition by nivolumab and CTLA-4 inhibition by ipilimumab in platinum-resistant ovarian cancer. The study involved 100 patients with recurrent ovarian cancer 12 months after the last platinum therapy. The patients qualified for the study had no more than three previous chemotherapy cycles for treatment of epithelial ovarian cancer, fallopian tube carcinoma, or primary peritoneal cancer. The chemotherapy scheme was based on platinum applied after surgical or non-surgical assessment of the stage of the disease. The results of this study have not yet been published (NCT02498600). Currently, there are two phase II trials of avelumab for ovarian cancer—one for front-line therapy in combination with carboplatin and paclitaxel (Javelin ovarian 100) and the other for recurrent platinum-resistant disease (Javelin ovarian 200). The Javelin ovarian 200 is the phase III trial of PD-1 inhibitor in patients with platinum-resistant, recurrent ovarian, fallopian tube, or peritoneal cancer. This three-arm trial compared avelumab alone or in combination with doxorubicin versus doxorubicin alone. The results of this study indicate that avelumab applied as monotherapy or in combination with doxorubicin can improve PFS or OS compared with standard chemotherapy in patients with platinum-resistant ovarian cancer [65].

The Javelin ovarian 100 (NCT022718417) is an ongoing phase III trial in patients with ovarian cancer. This three-arm study compares the efficacy of avelumab in combination with platinum-based chemotherapy, avelumab alone after platinum based-chemotherapy, and platinum-based chemotherapy alone.

The substantial efficacy of combination therapy of immune checkpoint inhibitors with bevacizumab is found in kidney cancer [66]. Bevacizumab is a recombinant monoclonal antibody that binds to the vascular endothelial growth factor (VEGF). As a result, the interaction of VEGF with Flt-1 (VEGFR-1) and KDR (VEGFR-2) receptors on the surface of the endothelial cells is inhibited, which leads to the reduction of vascularization and inhibition of cancer growth. Currently, atezolizumab is assessed in combination with paclitaxel, carboplatinum, and bevacizumab in patients with ovarian cancer stage III or IV, fallopian tube, or primary peritoneal cancer (IMagyn050, NCT03038100).

Another currently ongoing trial assessing many combination therapies in ovarian cancer is ATALANTE (NCT0289182). This study is a randomized, multisite phase III study assessing the efficacy of atezolizumab in combination with platinum-based chemotherapy and bevacizumab vs. placebo + platinum-based chemotherapy + bevacizumab. The study involved patients with recurrent platinum-sensitive ovarian cancer >6 months after the last platinum therapy [67].

The efficacy of immune checkpoint inhibitors and VEGF inhibitor therapy is assessed also in combination with other chemotherapy schemes in cases of platinum-resistant ovarian cancer. The multisite, randomized phase III trial is ongoing, assessing the safety and efficacy of atezolizumab with bevacizumab and chemotherapy (paclitaxel + doxorubicin) in comparison with placebo with bevacizumab and chemotherapy (paclitaxel + doxorubicin) in patients with recurring ovarian cancer, fallopian tube, or primary peritoneal cancer with one or two recurrences during 6 months after platinum-based chemotherapy or three recurrences.

#### 2.2.2. Combination Therapy with PARP Inhibitors

Currently there is an ongoing phase I/II study of durvalumab (NCT02484404) in combination with either the PARP inhibitor (olaparib) or the VEGFR inhibitor (cediranib). There was one partial response in nine evaluable ovarian cancer patients lasting >6 months with the combination of durvalumab and olaparib, and one partial response in five evaluable ovarian cancer patients treated with durvalumab and cediranib [68].

The ATHENA study (NCT03522246) is a randomized, multinational, phase III clinical trial involving patients with ovarian cancer. This study assesses rucaparib and nivolumab as a supportive treatment after the response for first platinum-based chemotherapy in newly diagnosed patients with ovarian cancer stage III and IV. ATHENA is composed of four arms covering obejmujących (the combination of rucaparib and nivolumab), rucaparib and placebo, nivolumab + placebo, and placebo only. ATHENA is one of the several studies that are currently assessing the potential of combining immune checkpoint inhibitor and PARP inhibitor: ARIES NCT03824704 (rucaparib + nivolumab), MEDIOLA (durvalumab + olaparib), and TOPACIO (pembrolizumab + niraparib).

#### 2.2.3. Future Possibilities: Vaccines

Vaccines are a promising therapeutic option in ovarian cancer. Scientists are focused on antigens having immunological potential, such as p53, CA125, MUC1, CEA, folic acid receptor alpha, NY-ESO-1, and proteins binding insulin growth factor. Many studies also concern the use of dendritic cells because of their strong antigen-presenting ability.

In a study published by Tanyi et al., the vaccine, generated by autologous dendritic cells (DCs) and pulsed with oxidized autologous whole-tumor cell lysate (OCDC), was injected intranodally in platinum-treated, immunotherapy-naïve, recurrent ovarian cancer patients. OCDC was administered alone (5 patients), in combination with bevacizumab (10 patients), or bevacizumab + low-dose intravenous cyclophosphamide (10 patients) until disease progression or vaccine exhaustion. During vaccine administration (392 doses in total), serious adverse effects were not observed. The increased immunological response of T cells against autologic cancer antigen was observed, which was associated with the increased survival [69].

The TPIV200 vaccine is directed against folic acid receptor alpha. Kalii et al. published the results of a phase I study evaluating the safety and efficacy of the vaccine in 14 patients with ovarian cancer stage II–IV (including primary ovarian cancer and fallopian tube cancer), which finished standard treatment at least 90 days before registration and without the recurrence of the disease. The vaccine was well-tolerated and did not show toxicity ≥3 stage. The induced or increased immune response in more than 90% of studied patients was observed. The response of T cells against FR developed slowly over the course of vaccination, with a median time to maximal immunity at 5 months. Despite slow development of immunity, responsiveness appeared to persist for at least 12 months [70]. The safety and immunogenicity of the combination of peptide vaccine targeting WT1 with nivolumab were evaluated in phase I study in patients with recurrent, platinum-treated ovarian cancer as a supportive treatment in second or third remission. In one year, the PFS rate was 64% [71].

### 2.3. Endometrial Cancer

In 2015, Le et. al published a phase II study showing the efficacy of pembrolizumab in tumors with a mismatch-repair deficiency (MMR). The RR rate and PFS rates were 71% and 67%, respectively, for MMR-deficient colorectal cancer patients (with two patients with endometrial cancer) [72]. In the KEYNOTE-028 phase I basket trial, patients with endometrial cancer were subjected to pembrolizumab therapy. This trial investigated the safety and efficacy of pembrolizumab in solid cancers with PDL1 positivity. In this study, 36 (48%) of 75 women with recurrent or progressive endometrial cancer were found to have PD1 positivity, and 24 were enrolled and analyzed in the endometrial cohort [73]. Qualified patients were treated with intravenous pembrolizumab 10 mg/kg every 2 weeks for a maximum of 24 months. The obtained results were promising, with a 13% ORR, 1.8 month median PFS at the data cut off, 19% 6 month and 14.3% 12 month PFS rates.

The preliminary results of another ongoing phase II trial of pembrolizumab showed that of nine women with MMR-deficient, persistent, or recurrent endometrial cancer, 56% had an ORR (n ¼ 5), including one complete response and four partial responses. There were no reported toxicities greater than grade 3 [74]. Recently obtained knowledge about the microenvironment in endometrial cancer and from preliminary immunotherapy trials that enrolled endometrial cancer patients encourages further attempts at immunomodulation in the treatment of aggressive forms of this disease [75].

In a study published by Santin et al., nivolumab was administered to two patients with endometrial cancer. First, patients with a mixed clear cell and endometrioid (CC/EAC) endometrial cancer, stage IIIA, after surgical treatment, chemotherapy (cisplatin/adriamycin/paclitaxel for 7 cycles), and vaginal cuff radiation, were given nivolumab 3 mg/kg biweekly. The second patient with a recurrent/metastatic highly aggressive variant of type II endometrial cancer, after a robotic-assisted surgical treatment and six cycles of docetaxel and carboplatin and brachytherapy, was given nivolumab 3 mg/kg every 2 weeks. Both patients showed good clinical response to treatment with adverse effects not exceeding grade 3 [76].

A phase Ia clinical trial of atezolizumab in patients with advanced endometrial cancer showed that atezolizumab has antitumor activity. In this study, 15 patients were treated with 15 mg/m^2^ atezolizumab in monotherapy every 3 weeks. The best ORR was 13%, a median PFS of 1.7 months, and a median OS of 9.6 months [77].

Avelumab was also tested in a recent phase II trial. In this ongoing trial, 16 patients received avelumab 10 mg/kg every 2 weeks until disease progression or unacceptable toxicity. The results of this study have not yet been published [78].

#### Combination Therapies

Mekkel et al. presented the results of a multicenter, open-label, single-arm, phase II trial (NCT02501096). The study involved 54 patients with advanced, metastatic endometrial cancer, who were subjected to no more than two systemic therapies. In this study, the efficacy of pembrolizumab in combination with lenvatinib (a multikinase inhibitor with antiangiogenic activity) was evaluated. Pembrolizumab was administered intravenously at a dose of 200 mg every 3 weeks, and lenvantinib was administered orally at a dose of 20 mg daily. The mean follow-up time was 13.3 months. In this study, 21 patients (39.6%) showed good clinical response during 24 weeks. The use of a combination of two drugs was well tolerated, but serious adverse events occurred in 16 patients (30%). The most common adverse events were hypertension (58%), fatigue (55%), diarrhoea (51%), and hypothyroidism (47%). The adverse events were similar to those occurring in a case of administration of these drugs as monotherapy [79]. Because of the good safety profile, a phase III clinical trial assessing combination therapy of pembrolizumab and lenvantinib is being planned.

Currently, there is an ongoing phase II study (NCT02549209) of carboplatin/paclitaxel and pembrolizumab in advanced, recurrent endometrial cancer; a randomized phase III study of carboplatin/paclitaxel with or without pembrolizumab (NRG-GYO18) in stage III/IV or recurrent endometrial cancer. There is an ongoing phase II study (NCT03015129) of durvalumab + tremelimumab and durvalumab alone in recurrent endometrial cancer with MMR status [80,81].

### 2.4. Side-Effects Associated with Immune Checkpoint Inhibitor Therapy

Despite the clinical benefits of immune checkpoint inhibitor therapy, undesirable side effects can occur during the treatment. Because the immune checkpoint inhibitors are not directed solely to tumor-specific T cells, these drugs, apart from induction of the desirable anti-tumor immune response, may cause the unintended activation of non-tumor-specific immune responses that target self-antigens expressed on healthy tissue. Indeed, in a variety of organs following immune checkpoint inhibitor therapy, immune related adverse events (irAEs) have been reported to occur. Such irAEs most frequently result in dermatological conditions, such as pruritis and mucositis (up to 68% of patients on anti-CTLA-4 therapy) [82]. Other common irAEs include gastrointestinal distress in the form of diarrhea and immune-mediated colitis, occurring in as high as 40% of ipilimumab-treated patients. Less common irAEs include hepatotoxicity, endocrinopathies, and pneumonitis. In rare cases, renal toxicity, neurotoxicity, cardiovascular toxicity, pancreatitis, hematological abnormalities, and ocular manifestations have been reported [82].

## 3. Conclusions

The results of the clinical studies show that immune checkpoint inhibitors, used as monotherapy or in combination with other immune checkpoint inhibitors or chemotherapy, are promising approaches in the treatment of gynecological cancers. Future studies may lead to the development of efficient therapy for gynecological cancer patients.

## Figures and Tables

**Figure 1 ijms-20-04705-f001:**
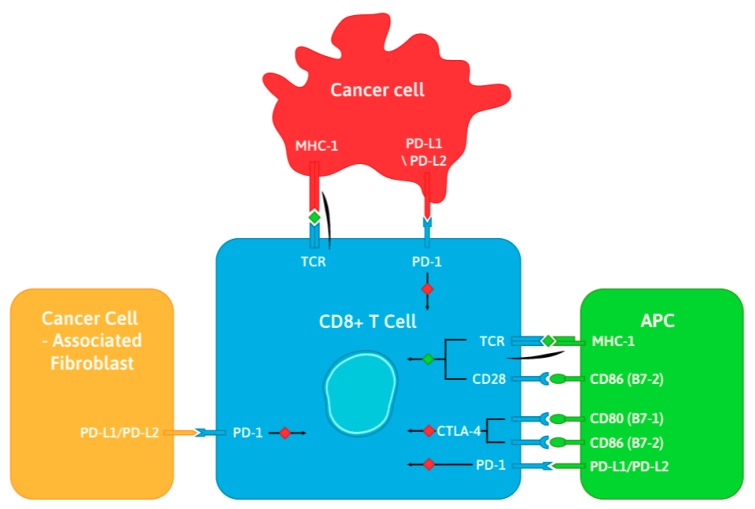
Mechanism of action of cytotoxic T lymphocyte-associated protein-4 (CTLA-4) and programmed cell death protein-1 (PD-1). CTLA-4 binds CD80 and CD86 proteins on antigen-presenting cells (APC), transmitting an inhibitory signal to T cells. PD-1 binds programmed death-ligand 1 and 2 (PD-L1/PD-L2) ligands expressed by APC, cancer cells, and cancer cell-associated fibroblasts, suppressing T cell inflammatory activity.

**Table 1 ijms-20-04705-t001:** Ongoing clinical trials with immune checkpoint inhibitors in gynecologic cancers.

Gynecologic Cancer	Clinical Trial	Intervention	Study Population	Phase
Cervical cancer	NCT02628067 KEYNOTE 158	Pembrolizumab	Advanced solid tumors	II
	NCT02488759 CheckMate 358	Arm 1: neoadjuvant/metastatic nivolumab Arm 2: nivolumab + ipililumab Arm 3: nivolumab + BMS-986016 Arm 4: nivolumab + daratumumab	Squamous cell carcinomas of the cervix, vulva, and vagina, plus other virus-associated malignancies	I/II
	NCT01711515	Primary chemoradiation followed by ipilumumab	Advanced cervical cancer stage IB-IIB with positive PA nodes only and stage IIB/IIIB/IVA with positive nodes	I
	NCT02635360	Arm 1: chemoradiation followed by pembrolizumab Arm 2: chemoradiation with concurrent pembrolizumab	Locally advanced cervical cancer	
	NCT02866006	BVAC-C vaccine	Metastatic, progressive, or recurrent HPV 16/18 cervical cancer after failed standard therapy	I
	NCT02128126	ISA101/ISA101b vaccine	Advanced, metastatic, or recurrent cervical cancer and HPV16positive	I/II
Endometrial cancer	NCT02549209	Pembrolizumab + carboplatin + paclitaxel	Stage III/IV or recurrent endometrial cancer	II
	NCT02899793	Pembrolizumab	Recurrent endometrial cancer	II
	NCT02982486	Nivolumab + ipilimumab	Non-resectable/metastatic sarcoma or high-grade endometrial cancer with MSI	II
Ovarian cancer	NCT02580058 JAVELIN Ovarian 200	Arm 1: avelumab Arm 2: avelumab + doxil Arm 3: doxil	Platinum-resistant/refractory EOC	III
	NCT02839707	Arm 1: doxil + atezolizumab Arm 2: doxil + atezolizumab + bevacizumab Arm 3: doxil + bevacizumab	Platinum-resistant EOC	II/III
	NCT02440425	Paclitaxel + pembrolizumab	Platinum-resistant EOC	II
	NCT02608684 PemCiGem	Pembrolizumab + gemcitabine + cisplatin	Platinum-resistant EOC	II
	NCT02891824 ATALANTE	Arm 1: placebo + bevacizumab + platinum chemo Arm 2: atezolizumab + bevacizumab + platinum chemo	Recurrent platinum-sensitive EOC	III
	NCT01928394 CheckMate 032	Arm 1: nivolumab Arm 2: nivolumab + ipilimumab Arm 3: nivolumab + ipilimumab + cobimetinib	Advanced or metastatic solid tumors	I/II
	NCT02498600	Arm 1: nivolumab + nivolumab maintenance Arm 2: nivolumab + ipilimumab + nivolumab maintenance	Recurrent or persistent EOC	II
	NCT03026062	Arm 1: sequential tremelimumab followed by durvalumab Arm 2: combination tremelimumab + durvalumab	Platinum-resistant and platinum refractory EOC	II
	NCT02726997	Durvalumab + carboplatin + paclitaxel	Advanced EOC with no prior treatment	I/II
	NCT02520154	Neoadjuvant carboplatin + paclitaxel followed by interval TRS and adjuvant carboplatin + paclitaxel + pembrolizumab	Advanced EOC with no prior treatment	II
	NCT02834975	Neoadjuvant pembrolizumab + carboplatin + paclitaxel following by interval TRS and adjuvant pembrolizumab + carboplatin + paclitaxel	Advanced EOC with no prior treatment	II
	NCT03038100 IMagyn050	Arm 1: carboplatin + paclitaxel + bevacizumab + atezolizumab Arm 2: carboplatin + paclitaxel + bevacizumab + placebo	EOC with no prior treatment	III
	NCT02718417 JAVELIN OVARIAN 100	Arm 1: carboplatin + paclitaxel Arm 2: carboplatin + paclitaxel + avelumab maintenance Arm 3: carboplatin + paclitaxel + avelumab + avelumab maintanence	Advanced EOC with no prior treatment	III

HPV: human papillomavirus, EOC: epithelial ovarian cancer, MSI: microsatellite instability.

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
