# Peer review of "Current Possibilities of Gynecologic Cancer Treatment with the Use of Immune Checkpoint Inhibitors"

_ijms, 2019, doi:10.3390/ijms20194705_

Round 1
Reviewer 1 Report
Please provides schemes/ illustrations for mechanism of action of PD-1. PDL-1 and CTLA-4 Please provide table/s for approved clinical trials with checkpoint inhibitors on different cancer types with particular emphasis on cervical cancers. Please list common side-effects of checkpoint therapy treatments in the discussion section.Author Response
Reviewer #1
Please provide schemes/ illustrations for mechanism of action of PD-1, PD-L1 and CTLA-4The scheme showing mechanism of action of PD-1, PD-L1 and CTLA-4 has been provided (Figure 1).
Please provide table/s for approved clinical trials with checkpoint inhibitors on different cancer types with particular emphasis on cervical cancersThe table presenting the ongoing approved clinical trials with the immune checkpoint inhibitors in cervical, endometrial and ovarian cancers has been provided (Table 1).
Please list common side-effects of checkpoint therapy treatments in the discussion sectionNew sub-section has been prepared describing common side-effects of the immune checkpoint inhibitors therapy (Section 2.4).
Reviewer 2 Report
This is comprehensive review article on a current important topic in cancer research for GYN cancers. Although it is not original it provides a good overview for those interested in the topic.
I have 2 comments however,
In the beginning the cost of cancer care is cited. Given the expensive costs associated with immunotherapy, I would delete the statements in the introduction regarding cost.2. Line 45, change "Nowadays", to "Currently"
Author Response
Reviewer #2
In the beginning the cost of cancer care is cited. Given the expensive costs associated with immunotherapy, I would delete the statements in the introduction regarding costThe statement in the introduction about the costs of cancer care has been deleted.
Line 45, change "Nowadays", to "Currently"The word „Nowadays” has been changed to „Currently”.